# Prognostic Scores in Acute Liver Failure Due to Viral Hepatitis

**DOI:** 10.3390/diagnostics13061035

**Published:** 2023-03-08

**Authors:** Sagnik Biswas, Ramesh Kumar, Subrat Kumar Acharya

**Affiliations:** 1Department of Gastroenterology and Human Nutrition, All India Institute of Medical Sciences, New Delhi 110029, India; 2Department of Gastroenterology, All India Institute of Medical Sciences, Bihar 801507, India; 3Fortis Escorts Digestive and Liver Institute, Okhla, New Delhi 110025, India

**Keywords:** ALF, liver transplant, plasma exchange, extracorporeal liver support, virus

## Abstract

Viral infections are among the major causes of acute liver failure (ALF) worldwide. While the role of agents such as hepatitis A, B, C, D and E viruses in precipitating ALF are well known, improvements in serological assays have led to the detection of viral agents such as Epstein Barr virus, cytomegalovirus etc. as atypical causes of ALF. Despite the plethora of literature available on viral hepatitis and ALF, there is very limited large-scale epidemiologic data on the prevalence, risk factors of progression and outcomes in ALF of viral causes. This is important as viral infections remain the leading cause of ALF in the East and in developing countries, while the impact of viral ALF in the West has largely been ameliorated by effective vaccination and sanitization programs. This review focuses specifically on the available prognostic scores that aid in the management of ALF of viral etiologies while also briefly reviewing the current literature on newer viral agents known to cause ALF, risk factors of progression, outcomes and how management algorithms can be developed by incorporation of prognostic scoring systems for referral and transplant listing.

## 1. Introduction

Acute liver failure (ALF) is clinically defined as coagulopathy (internationalized normalized ratio (INR) > 1.5) and encephalopathy following an acute hepatic insult in a patient without pre-existing liver disease [1]. The time interval between the onset of jaundice and encephalopathy varies from 4 weeks in India to 26 weeks in the United States [2]. It is considered a medical emergency and is associated with high mortality rates of 50–75% [3]. The etiology of ALF is myriad, varies globally and includes drugs, infectious etiologies, such as viruses, and rare metabolic diseases such as Wilson’s disease. In the West, most ALF cases are due to drugs and toxins with acetaminophen being the most prevalent. By contrast, in the East and the developing world, ALF is mainly due to viral infections. The common etiologies of ALF are provided in Table 1.

ALF is considered a success story in gastroenterology as the introduction of liver transplantation (LT) in its management has proven to be a definitive therapy for afflicted patients, and LT has decreased mortality rates to as low as 20% [4,5]. ALF patients are currently being given the highest priority for LT (“Status 1A”) by the United Network for Organ Sharing (UNOS), with median wait times as low as 48 h [6]. The introduction and widespread usage of live donor liver transplants has further reduced the waitlist times while also providing a potential solution to the ever-increasing disparity between organ donors and recipients [7]. In parallel with the developments in LT, there have been better understanding of pathophysiology and improvements in critical care medicine, which have led to improved transplant-free survival (TFS) rates [8].

Thus, it is increasingly being recognized that not all ALF patients require LT and there is a need for objective measures to identify those who do need it [9]. This subset of patients would benefit from early referral to specialized liver units (SLU) and transplant listing. Several prognostic scores exist which serve as prognostic models to identify patients with poor outcomes who would require LT [10]. The current article reviews the various prognostic models developed for patients with ALF due to a viral etiology, along with their performance characteristics, applicability and limitations.

## 2. Viral Etiologies of ALF

Viruses are one of the most common causes of ALF worldwide. Several viral agents are known to cause ALF, which may be classified as follows (Table 2):

## 3. Epidemiology

Despite the knowledge that several viruses may cause ALF, there is limited global epidemiological data on viral-ALF [11]. Some of the reasons behind this are: (i) recent identification of certain viruses as causative agents of liver diseases, (ii) development of better diagnostic assays which allowed for diagnosis of atypical viral agents causing ALF (which may have been classified as idiopathic ALF), (iii) the inconsistent implementation of vaccination programs for preventable viral diseases across countries. As a result of this, one finds that the burden of hepatitis A and B-related ALF is very high in Southeast Asia due to poor penetration of vaccination programs as compared to the West, where common etiologies of ALF are drugs, toxins and metabolic diseases. Patterson et al. reported that cases of hepatitis B-related ALF (HBV-ALF) in Europe have decreased to 19% following routine immunization, while similar changes in hepatitis A related ALF (HAV-ALF) have been seen in Argentina, with a decrease in incidence to 25% [12]. Similarly, hepatitis E virus-related ALF (HEV-ALF) was reported to have a point prevalence of 32% (range: 3–70%) in this review. The hepatitis E vaccine (Hecolin) was introduced in China in the year 2012 but is licensed by the WHO for use only in outbreaks. Despite being introduced over a decade, this vaccine has not been incorporated in national programs, but may potentially decrease the prevalence of HEV infections in endemic areas if approved [13]. The importance of knowing the epidemiology of viral ALF extends beyond the design of national programs and preventive healthcare. Multiple groups have reported that ALF most commonly presents in the fourth decade of life and is more prevalent in women [14,15,16]. Thus, most affected patients are relatively young and at a risk for significant morbidity and mortality. However, mortality rates vary in different countries owing to the availability of donor livers, the prevalent etiology of ALF and referral practices. The survival rates in ALF with expectant management range between 50 and 60%, which may improve to 80% after liver transplant (LT) [1]. Thus, 20% of patients with ALF will succumb to the illness despite receiving definitive care with LT. There are multiple reasons why an LT does not ensure survival in all ALF patients, which include: (i) incorrect timing of liver transplant (pre-transplant waiting >5 days), (ii) post-transplant infections, (iii) recipients’ characteristics (such as higher age, higher BMI, concomitant renal impairment, vasopressor requirement), (iv) deficiencies in referral systems with lack of SLU and (v) procedural morbidity and mortality associated with the transplant procedure.

The timing of the LT is a very pertinent but unanswered question in the management of ALF, despite the immense research and literature available on the subject. Current guidelines advocate that all patients should be considered for LT if they deteriorate clinically and should be referred to an SLU for specialized care while awaiting a transplant [17,18]. The impact of referral to an SLU in improving outcomes in these patients has also been noted in recent studies [19,20]. Several procedures that allow us to sustain a patient clinically have also become available, such as plasma exchange (PE) and an extracorporeal liver support (ECLS) device, which has shown some benefit in improving biochemical and clinical parameters (ECLS) and in survival rates (PE) in randomized trials [21,22]. Thus, when faced with a patient presenting with ALF, the clinician should be able to answer the following questions: (i) whether the patient require a liver transplant or expectant management; (ii) how to objectively identify parameters which would enable to identify those in need of LT; and (iii) how frequently these objective parameters should be reassessed in a patient. Prognostic scoring systems can help answer these in ALF patients. Several such prognostic systems are available, each with its benefits and shortcomings, which are discussed in this review article with respect to their use in viral-ALF.

## 4. Importance of Prognostic Systems

Prognostic systems can help objectively identify patients requiring an LT. To most efficiently manage such patients, the prognostic system should have the following properties [23]:(i)**Dynamicity**

ALF is a dynamic condition which requires close monitoring for early identification of poor prognostic parameters. A delay in identification may lead to complications such as infections, cerebral edema etc., significantly decreasing survival in these patients. Thus, the prognostic score should also reflect the changing prognosis of patients with changing clinical and biochemical parameters [24].

(ii)
**Applicability**


We have highlighted that a multitude of factors can cause ALF. An ideal prognostic system should be able to predict outcomes across etiologies of ALF. For example, high bilirubin values in anti-tubercular drug therapy-induced ALF are an independent risk factor determining poor outcomes, while it is not very relevant in patients with acetaminophen-induced ALF (APAP-ALF) [25]. An ideal score should thus be universally applicable and provide accurate results irrespective of etiology.

(iii)
**Accuracy**


The accuracy of a prognostic score in ALF would be its ability to identify those who require an LT. Thus, the test would be required to have a high sensitivity and specificity in order to increase true positives and eliminate false positives). It would also need to have a high positive predictive value (PPV) so that those who require a transplant are identified by the test, as well as a high negative predictive value (NPV) to ensure those who would survive without a transplant do not receive an organ. Unfortunately, no single prognostic score meets all the criteria mentioned above [9,26].

(iv)
**Ease of use**


The most commonly used clinical prognostic scores are those which are the least cumbersome with the highest yields. This is one of the major drawbacks of scores with multiple parameters (such as the acute physiology and chronic health evaluation (APACHE) score), which have multiple components and are thus cumbersome. Similarly, the components of the test should be commonly available. Several newer tests use serological markers that are not available at most centers, such as M30 and M65 (circulating apoptotic markers), limiting their use to a research setting or academic centers [27,28].

## 5. Clinical Presentation of Viral ALF

Clinically, the natural history of hepatitis caused by the hepatotropic viruses (HAV, HBV, HCV, HDV and HEV) is well known [29]. The mode of infection, incubation period and outcomes of acute viral hepatitis are provided in Table 3. There are limited data available on factors which predict the progression from viral hepatitis to ALF. In general, ALF is classified based on the time interval between the appearance of jaundice and the onset of encephalopathy [30]. This also has prognostic significance as a more rapid onset of encephalopathy (hyperacute or acute presentation) has better outcomes than a more subacute or delayed presentation. However, there is substantial variability in defining ALF based on icterus-encephalopathy interval. The proposed different classification systems used for ALF are provided in Table 4.

## 6. Approach to Management

Ideally, all patients with ALF should receive care in an intensive care unit [43]. ALF is a dynamic condition where stringent monitoring is required to prevent complications which would adversely affect the outcome [42]. One of the most feared outcomes is cerebral edema which may lead to a progressive and irreversible decline in neurological functions and render a patient unfit for liver transplantation [44]. Thus, the primary target of management in ALF is good triaging (to identify patients with poor prognosis at baseline) and early referral to higher centers with SLU to be considered for early transplant listing or other bridging methods such as plasmapheresis or artificial liver support systems (ALSS), as may be relevant [45]. In this regard the clinical judgment of the physician is aided by several objective prognostic scores which have been detailed below. Some of these prognostic scores are applicable to ALF of any etiology (e.g., Clichy score, King’s college criteria (KCC) for non-acetaminophen ALF (non-APAP ALF) or model for end-stage liver disease (MELD) score) while some scoring systems are specific to the precipitating etiology (e.g., hepatitis A related-ALF (ALFA) score). These scoring systems have predominantly been derived from a retrospective analysis of ALF cohorts and prospectively validated. Each scoring system has its own benefits and drawbacks, but most have not been compared with each other in randomized trials; hence, there is no single best scoring system which is universally accepted. Similarly, when patients are considered for LT in cases of ALF, no single scoring system is relied upon and it is a composite decision based on clinical and objective parameters [4].

## 7. Prognostic Scoring Systems in Viral ALF

There are several prognostic systems available for use in ALF. As viral ALF is the leading cause of ALF in the East, most of these systems have been used for prognostication in viral ALF along with other etiologies. For this review, we have limited the discussion to scoring systems which have either been used in viral ALF or have been developed for specific viral etiologies. As noted from the data provided in Table 5, very few scoring systems have been developed exclusively for the prognosis of viral hepatitis. Further, the number of patients of viral ALF included in the derivation cohort of these scoring systems was limited. The etiology-specific scoring systems are largely limited to hepatitis A and E. The details of the various cut-off values of prognostic scoring systems in predicting outcomes are shown in Table 6. The scoring systems for hepatitis B virus largely relate to acute-on-chronic liver failure, while scoring systems for hepatitis C, hepatitis D and the other less commonly encountered viruses are not available [46,47].

## 8. Management of Viral ALF

Viral ALF can be precipitated by multiple agents, yet only a few have specific antiviral agents which may be used—hepatitis B (entecavir, tenofovir), HSV (acyclovir) and CMV (ganciclovir, valganciclovir) [74,75,76]. Ongoing research has highlighted the unique role of host factors such as very low density lipoproteins, low density lipoproteins, high density lipoproteins and apolipoproteins in mediating viral entry of HCV through suppression of the transforming growth factor beta pathway. Identification of such newer pathways not only provides us clearer understanding of pathways of viral replication and infection, but also provides newer targets for drug therapy. Better understanding of these interactions may also lead to identification of newer risk factors which predict progression of disease from hepatitis to liver failure [77,78]. Novel drug targets are being identified to combat entry of hepatotropic viruses into cells and replication such as nicotinamide in hepatitis A [79], recombinant HEV virion [80] and the newly introduced bepirovirsen in hepatitis B [81], although these are yet to be tested in acute liver failure. Emerging data from animal studies are also available on immunotherapy with agents such as thymosin alpha 1 [82], mesenchymal stem cells and their exosomes in ALF [83] as immunomodulators to prevent immune dysfunction in ALF. If successful, they would help decrease the burden on the organ pool and contribute to improving transplant free survival rates in ALF further. As the atypical viral infections that precipitate ALF most commonly occur in the immunocompromised, decreasing the doses of immunosuppressive drugs may prevent progression from acute viral hepatitis to ALF and thus drug therapy could be reserved for advanced or severe cases [84].

Once a patient progresses to ALF, the management protocol of the patient becomes standardized, as shown in Figure 1. Ideally, all ALF patients should receive care in an intensive care unit (ICU) or SLU which has been shown to improve survival in ALF patients in multiple studies [20,85]. The key decisions in management are to triage patients who need a transplant and to time the transplant correctly so as to maximize the chances of recovery and survival [4]. While clinical assessment is subjective and may show interobserver variations, objective prognostic scores form the backbone of decision-making in ALF. Ideally the prognostic scores are calculated at admission to identify patients with poor prognosis on entry to the system. Patients are resuscitated and started on supportive care. These scores are then repeated at various intervals to assess the progression or resolution of the underlying condition. Thus, we can objectively assess those who will recover with supportive care only (lower priority for transplant listing or delisting) and those who are clinically worsening and thus need to be listed for an LT [86].

Current issues in the clinical management deal with the choice of score to be used, as there is no single “best scoring system” available, and the frequency at which these should be repeated. These practices are largely adapted to the centers at which the patient is admitted. For example, a high-volume transplant center would prefer more frequent reassessments (6–12 hourly) to identify early features of deterioration and proceed for an LT. At present, there is no consensus on the frequency of repeat scoring and assessment. If the patient is found to deteriorate clinically or have poor prognosis as per the scoring systems, they should be listed for transplantation and referred to an SLU (if not already admitted to one). Over here the decision may be to proceed to a transplant based on the availability of a donor organ or utilize a bridge to transplant such as plasmapheresis or artificial liver support systems (ALSS) which would sustain the patient clinically to allow more time for obtaining the donor organ [22,45,87,88]. In this respect, plasmapheresis has been shown to improve both clinical parameters as well as short-term survival, whereas several ALSS have been shown to improve clinical and biochemical parameters (bilirubin, hepatic encephalopathy) without impacting survival [22,89].

Advances in liver transplant such as availability of live donor liver transplants (LDLT), expanded criteria/marginal liver and hepatocyte transplants have changed the landscape of ALF management [90,91]. The availability of LDLT significantly reduces the waiting time for a donor organ for a recipient. LDLT is also associated with less cold ischemia time (CIT), although this does not translate to a clinically lower incidence of biliary strictures or lower rejection rates [92]. LDLT is thus a popular option in countries such as India where organ donation after death is not a popular option and thus a small donor pool is available for dead donor liver transplant (DDLT) [93]. Post-transplant survival rates in ALF are provided in Table 3 and vary as per the etiology. Overall, the TFS is best for hepatitis E while the 1- and 5-year post transplant survival is excellent for HBV-ALF.

## 9. Conclusions

A large number of viral agents, both typical and atypical, can cause acute liver failure. Over the past three decades, improvements in critical care management and a better knowledge of pathophysiology have led to a dramatic rise in the transplant free survival rate for viral-induced acute liver failure patients. However, a significant number of patients still need a liver transplant and the post-transplant survival rate has improved to about 80%. Several prognostic scoring systems exist for prognostication and the decision to proceed with liver transplantation in such patients. However, the lack of reliable prognostic models frequently makes it difficult to identify transplant candidates early and appropriately, necessitating further research to produce a more reliable and widely used prognostic model for viral-induced acute liver failure.

## Figures and Tables

**Figure 1 diagnostics-13-01035-f001:**
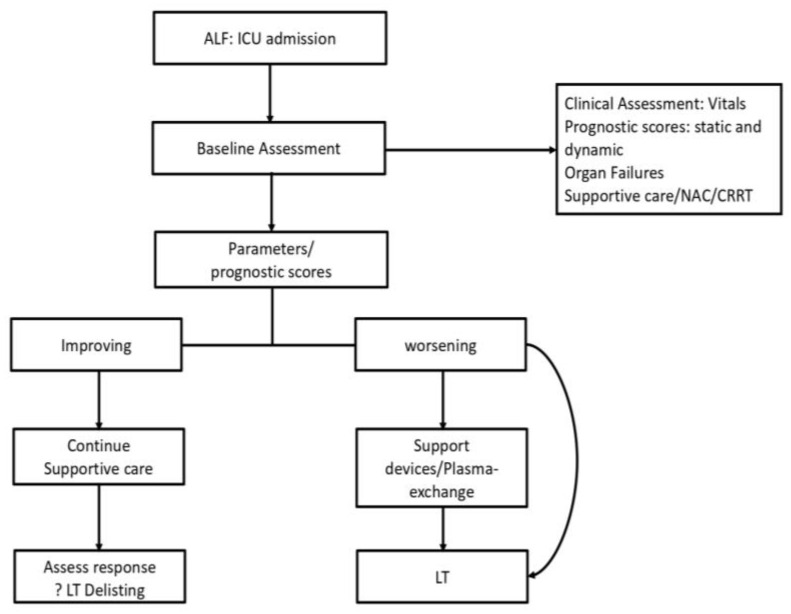
Suggested management algorithm for ALF.

**Table 1 diagnostics-13-01035-t001:** Etiologies of acute liver failure [1,3].

Etiologies	Examples
Viral Hepatitis	Hepatitis A, B, C, D, E, cytomegalovirus, Epstein–Barr virus, herpes simplex, varicella zoster, adenovirus, dengue virus
Drug-induced	Acetaminophen (APAP), isoniazid, ketoconazole, nitrofurantoin, rifampin, herbal medications
Autoimmune Hepatitis	-
Metabolic Disease	Wilson’s Disease
Vascular Diseases of Liver	Budd-Chiari Syndrome, veno-occlusive disease of the liver
Pregnancy-related liver failure	Acute fatty liver of pregnancy, pre-eclampsia
Malignant infiltration	Breast carcinoma, hematologic malignancies
Toxin exposure	Mushroom, rat poison, yellow phosphorus and other toxic agents
Miscellaneous	Partial hepatectomy, sepsis, hemophagocytic lymphohistiocytosis, hepatic ischemia

**Table 2 diagnostics-13-01035-t002:** Known viral agents which may cause acute liver failure [1,3].

(I) Based on the Availability of Effective Vaccine
Vaccine-preventable viruses
Hepatitis A
Hepatitis B
Viruses with no available vaccine
Hepatitis C
Hepatitis D
Hepatitis E
**(II) Based on the immune status of the patient**
Immune competent (although these may also occur in the immunocompromised)
Hepatitis A, B, C, D, E
Dengue virus
Immunocompromised
Herpes simplex virus (HSV-1 and HSV-2)
Cytomegalovirus (CMV)
Epstein–Barr Virus (EBV)

**Table 3 diagnostics-13-01035-t003:** Clinical outcomes of acute viral hepatitis-related acute liver failure.

Viral Agent	Mode of Infection	Incubation Period	Factors Leading to Progression to ALF	Transplant-Free Survival (TFS)	Post-Transplant Survival	
					1-year	5-years
Hepatitis A virus [29,31,32]	Feco-oral route	2–8 days	Old age, chronic viral hepatitis, underlying liver pathology	57–69%	69%	69%
Hepatitis B virus [29,31,33,34]	Parenteral route	4–26 weeks	Total bilirubin >5x upper limit of normal, HBe antigen negative status, concomitant alcohol use	25%	88%	85%
Hepatitis C virus [29]	Parenteral route	2–26 weeks	Uncommon	-	-	-
Hepatitis D [35]	Parenteral route	2–8 weeks	Unknown Possible HBV coinfection	-	-	-
Hepatitis E virus [36,37]	Feco-oral route	2–9 weeks	Poorly known causes	55.1%	-	-
CMV [38]	Contact through infected body fluids: blood, urine, saliva etc. Can be transmitted parenterally	3–12 weeks	Poorly known causes	-	-	-
HSV [39]	Contact with sores, saliva, surfaces in and around the mouth	2–12 days	Poorly known causes	20%	-	-
EBV [40]	Contact through infected body fluids: blood, urine, saliva etc. Can be transmitted parenterally	4–6 weeks	Primary or secondary immunosuppression	50%	100%	-
Dengue Virus [41]	Bite of the infected Aedes Aegypti mosquito	3–10 days	Young age ≤40 years, atypical lymphocytes >10%, platelets <50,000 per mm^3^	33.3%	-	-

**Table 4 diagnostics-13-01035-t004:** Classification of ALF based on the interval between onset of jaundice and encephalopathy.

O’Grady System [42]	Weeks from Jaundice to Encephalopathy
Hyperacute	0–1
Acute	1–4
Subacute	4–12
Bernuau System [42]	
Fulminant	0–2
Subfulminant	2–12
Japanese Classification (Mochida) [42]	
Fulminant	0–8 days
Acute	Within 10 days
Subacute	11 days to 8 weeks
Late-onset	8–12 weeks

**Table 5 diagnostics-13-01035-t005:** Currently available scoring systems with individual components and limitations.

Name	Component	Limitations
Based on the Severity of Liver Dysfunction		
MELD Score [48,49,50]	Serum Creatinine, Bilirubin, INR	-Predominantly retrospective analyses. -The use of MELD as a dynamic index has not been explored in large prospective studies. -Variable ideal discriminatory cut-off values -INR is subject to interlaboratory variation
Clichy Score [51,52]	Factor V levels with respect to age	-The etiology of viral ALF was predominantly hepatitis B. -Factor V is not a routinely available parameter
BiLE Score [53]	Bilirubin, lactate and etiology	-It is a retrospectively developed criteria. -The performance of the same was not validated prospectively. -The derivation cohort predominantly comprised of patients with ALF of indeterminate etiology
ALFED Score [24]	Arterial Ammonia, bilirubin, HE greater than Grade II, INR	Waiting time of 48 h may delay the selection of patients and referral for transplant.
ALFSG Score [54]	Coma grade, INR, bilirubin, phosphorus, and M_30_ levels	Major limiting factor is the use of cytokeratin-18 cleavage fragments, which are not routinely available at all centers.
ALF-OF score [55]	CLIF-C OF score Norepinephrine dose	Developed for patients with acetaminophen-induced ALF. Validation for viral ALF awaited.
Clinical Prognostic Indicator (CPI) score [49]	Age ≥ 50 years, JEI > 7 days, Grade 3 or 4 HE, cerebral edema, PT ≥ 35 s, and creatinine ≥ 1.5 mg/dL	Retrospective analysis No patient received an LT
**Etiology-Specific Scoring Systems**		
King’s College Criteria [49,56] (For non-APAP)	INR, age, etiology, duration of jaundice to encephalopathy, bilirubin	Low sensitivity implies that a large number of patients who require LT would be missed by the scoring system. Static score hence does not reflect the evolving nature of ALF clinically
Hepatitis A related ALF [57] (ALFA) score	Age, bilirubin, INR, ammonia, creatinine and hemoglobin	Model-based on retrospectively collected data. Single-time assessment based on values on the day of diagnosis of ALF (static score)
ALFSG Index for Hepatitis A [32]	Serum ALT < 2600 IU/L, creatinine > 2.0 mg/dL, need for mechanical ventilation and need for vasopressors	Limited numbers for derivation cohort
Prognostic Nomogram for Hepatitis E [58]	Gamma-glutamyl transpeptidase, albumin, total bilirubin, urea nitrogen, creatinine, international normalized ratio, and neutrophil-to-lymphocyte ratio	Prospective validation required in larger population samples
Scoring model of severe viral hepatitis (SMSVH) score [59]	Clinical type, hepatic encephalopathy, serum sodium and prothrombin activity	Chronic liver failure patients were included in the derivation cohort. No details on LT available. Needs validation
**Non-Liver related scores of organ dysfunction**		
Sequential Organ Failure (SOFA) Score [60,61]	P/F ratio, MAP/inotrope use, bilirubin, creatinine, platelets, GCS	Limitation of the derivation cohort to APAP-ALF Difficult to assess CNS involvement in intubated patients Not prospectively validated in viral ALF
Acute Physiology and Chronic Health Evaluation (APACHE) Score [61,62]	Multiple serologic and clinical markers	Limitations: score is cumbersome to calculate and is not validated for use. No additional benefit as compared to the MELD score or KCC.
**Stand-alone serological markers**		
Serum arterial ammonia [63,64]	-	Waiting for 72 h to assess persistent hyperammonemia may result in delayed referral of patients for LT. Ammonia levels can be influenced by non-hepatic factors
Serum phosphate levels [65,66]	-	Performs poorly compared to other markers, such as lactate, in predicting the outcome Limited assessment of outcomes specifically in viral ALF patients in prior studies
Blood lactate [67,68]	-	Lactate shows mixed results in non-acetaminophen related ALF, with studies advocating both for and against lactate’s utility as a prognostic tool. Needs further validation in viral ALF
Serum alpha-fetoprotein [69]	-	Needs further validation in viral ALF
**Research-based scores not commonly used clinically**		
Monocyte HLA-DR expression [70,71]	-	It is yet to be validated and is unavailable at most centers.
Serum Gc globulin [72,73]	-	The test is not readily available and needs validation in a larger cohort prior to use.

**Table 6 diagnostics-13-01035-t006:** Utility of prognostic scores in predicting mortality/outcome in patients with viral-ALF.

Prognostic Score	Cut-Offs	Sensitivity	Specificity	PPV	NPV	DA
Based on severity of liver dysfunction						
MELD score	≥35	86%	75%	88%	73%	83%
Clichy score	factor V level <20% in patients who were <30 years old Factor V level <30% in patients >30 years old	69%	50%	64%	55%	-
BiLE score	≥6.9	79%	84%	89%	71%	-
ALFED score	≥3	94%	59%	74%	90%	78%
	≥4	90%	80%	85%	87%	86%
	≥5	70%	93%	93%	71%	80%
ALFSG score		84.7%	59.2%	-	-	74.6%
ALF Organ Failure (ALF-OF) score	5.58	82.6%	89.5%	82.6%	89.5%	-
Clinical Prognostic Indicator (CPI) score	1	100%	9.6%	66.2%	100%	67.4%
	2	97.8%	42.3%	75%	91.7%	77.8%
	3	73.9%	86.5%	90.7%	65.2%	78.5%
	4	30.4%	100%	100%	44.8%	55.6%
Etiology-specific scoring systems						
King’s college criteria (KCC)	-	58.2%	100%	-	-	27.7%
ALFSG Index for Hepatitis A	≥1 Factor	100%	56%	65%	100%	-
	≥2 Factor	92%	88%	86%	93%	-
	≥3 Factor	62%	94%	89%	75%	-
	≥4 Factor	8%	100%	100%	57%	-
Scoring model of severe viral hepatitis (SMSVH)	5	77.7%	88.0%	-	-	-
Non-liver related scores of organ dysfunction						
Sequential Organ Failure (SOFA) Score	>6 by 72 h or >7 by 96 h	90%	69%	96.9%	98.8%	-
Acute Physiology and Chronic Health Evaluation (APACHE) Score	>15	82%	98%	-	-	-
Standalone serological markers						
Serum arterial ammonia	Baseline arterial ammonia > 124 mol/L	78.6%	76.3%	-	-	77.5%
Serum phosphate levels	Level of 1.2 mmol/L at 48 to 96 h after acetamenophen overdose	89%	100%	100%	98%	-
Blood lactate	Post-resuscitation arterial lactate cut-off 3.0 mmol/L	76%	97%	-	-	-
Serum alpha-fetoprotein	The ratio of AFP level on day 3 as compared to day 1 was >1 in 71% of survivors as compared to <1 in 80% of non-survivors.	-	-	-	-	-
Research based scores which are not commonly used clinically						
Monocyte HLA-DR expression	Monocyte HLA-DR expression 15% or less	96%	98%	-	-	100%
Serum Gc globulin	Cut-off level of 80 mg/L	49%	90%	85%	43%	-

## Data Availability

Not applicable.

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
