# Peer review of "Prognostic Scores in Acute Liver Failure Due to Viral Hepatitis"

_diagnostics, 2023, doi:10.3390/diagnostics13061035_

Round 1
Reviewer 1 Report
The article is devoted to risk factors of progression, outcomes of acute liver failure (ALF) and how management algorithms can be developed by incorporation of prognostic scoring systems.
I believe that the authors did not consider important data in the manuscript, for example:
1. Issues related to the mechanisms of ALF pathogenesis;
2. General clinical manifestations of ALF;
3. Are there any differences in the course of ALF caused by different reasons?
4. The frequency of occurrence of ALF of different etiologies was not given;
5. It is not clear why patients with Herpesvirus/Parvovirus/ Human parainfluenza viruses and others are immunocompromising;
6. Tables 1 and 2 are uninformative, you need to provide references.
For these reasons, the scientific value of the article is reduced.
Reviewer 2 Report
Biswas et al. reviewed that the prognostic scores in ALF due to viral hepatitis.
1. In abstract section, make a correction from “Hepatitis A,B,…” to “hepatitis A, B,…”.
2. In Table 2, “(I) Hepatitis E” section, authors should describe about “the recombinant hepatitis E vaccine (Hecolin), also designated HEV 239”. See: https://www.who.int/groups/global-advisory-committee-on-vaccine-safety/topics/hepatitis-e-vaccines#:~:text=The%20recombinant%20hepatitis%20E%20vaccine%20%28Hecolin%29%2C%20also%20designated,239%20assembles%20as%20homodimers%20resulting%20in%20virus-like%20particles.
3. In Table 2, “(II) Based on the immune status of the patient” part, describe the vaccination.
4. In Page 3, make a correction from “Hepatitis…” to “hepatitis…”.
5. Authors should describe the recent development of antivirals. See: Sasaki-Tanaka R, et al. J Virol. 2023 Feb 2:e0198722. doi: 10.1128/jvi.01987-22. PMID: 36728416; Yuen MF, et al. N Engl J Med. 2022 Nov 24;387(21):1957-1968. doi: 10.1056/NEJMoa2210027. PMID: 36346079; Primadharsini PP, et al. J Virol. 2022 Mar 23;96(6):e0190621. doi: 10.1128/jvi.01906-21. PMID: 35107380
Reviewer 3 Report
This review is in my opinion well written. A few times the Authors write (As written above, as already mentioned). I think such sentences should be removed.
In Hepatitis C infection there are several blood measurements that I think Authors should include in their Review. These measurements could also have importance for the several other viral liver infections that are discussed. Please include in your Tables these blood measurements: VLDL, LDL, apo E, apo E4, Apo A1, HDL. Also how these blood lipoproteins and lipids affect the progression to ALF and the prognosis of ALF.
Reviewer 4 Report
This paper is well conducted and shows the implication of the authors in acute liver failure.
In my opinion, I would add some comments regarding the use of potential biomarkers in ALF and immunomodulatory therapy and their correlation with LT
I do not find suitable the simulation with 3 patients. It was better to exemplify with patients from real practice if you believe the simulation is necessary
Regarding the references, I believe there are quite more self-citations.
Round 2
Reviewer 1 Report
The authors answered all my questions and comments and corrected the manuscript
Reviewer 4 Report
The new version of the manuscript is improved and I suggest it can be accepted in this new form